# A Study of Defect Detection Techniques for Metallographic Images

**DOI:** 10.3390/s20195593

**Published:** 2020-09-29

**Authors:** Wei-Hung Wu, Jen-Chun Lee, Yi-Ming Wang

**Affiliations:** 1Department of Mechatoronics Engineering, National Changhua University of Education, Changhua City 50007, Taiwan; d0551003@mail.ncue.edu.tw (W.-H.W.); wangym@cc.ncue.edu.tw (Y.-M.W.); 2Department of Telecommunication Engineering, National Kaohsiung University of Science and Technology, Kaohsiung City 80778, Taiwan

**Keywords:** metallographic analysis, deep learning, convolutional neural network, residual neural network

## Abstract

Metallography is the study of the structure of metals and alloys. Metallographic analysis can be regarded as a detection tool to assist in identifying a metal or alloy, to evaluate whether an alloy is processed correctly, to inspect multiple phases within a material, to locate and characterize imperfections such as voids or impurities, or to find the damaged areas of metallographic images. However, the defect detection of metallography is evaluated by human experts, and its automatic identification is still a challenge in almost every real solution. Deep learning has been applied to different problems in computer vision since the proposal of AlexNet in 2012. In this study, we propose a novel convolutional neural network architecture for metallographic analysis based on a modified residual neural network (ResNet). Multi-scale ResNet (M-ResNet), the modified method, improves efficiency by utilizing multi-scale operations for the accurate detection of objects of various sizes, especially small objects. The experimental results show that the proposed method yields an accuracy of 85.7% (mAP) in recognition performance, which is higher than existing methods. As a consequence, we propose a novel system for automatic defect detection as an application for metallographic analysis.

## 1. Introduction

Metallography is the study of the physical structure and components of metals through the examination of specimens with a metallurgical microscope. Microstructural characterization allows knowing the components of a metallographic in order to determine the material species and properties. In fact, a microstructure typically is a combination of different constituents, also known as phases, which produce complex substructures that store information related to the origin and formation mode of a material defining all its physical and chemical properties. Despite the dynamic development of digital photography and computer systems, metallographic analysis remains the task of experts who “manually” evaluate a given picture of the structure. In this case, defect detection of metallography appears extremely difficult, especially in the cases when there are mixtures of different phases with various substructures.

Since the success of the convolutional neural network (CNN) in pattern recognition, we have recently witnessed huge advances in object detection in the field of computer vision. Object recognition is crucial for autonomous cars, security, surveillance, and industrial applications which use deep learning methods such as region-based convolutional neural networks (R-CNN) [1], single-shot multibox detectors (SSD) [2], you only look once (YOLO) [3], and deep residual networks (ResNet) [4]. This motivation has led us to develop a system for metallographic analysis based on artificial neural networks, specifically on deep learning. In this study, we propose multi-scale Resnet (M-ResNet)—a novel ResNet architecture. This new architecture inherits the advantages of both ResNet and YOLOv4 [5]. It achieves state-of-the-art performance for defect detection in metallographic analysis. Experimental results validate the high accuracy of M-ResNet compared with other well-established baselines for defect detection on our datasets.

The remainder of the paper is organized as follows. Section 2 briefly introduces related works. A detailed description of the proposed strategy for object detection is given in Section 3. The experimental results and comparisons with other representative object detection methods are discussed in Section 4, and Section 5 gives the conclusion and proposes future work.

## 2. Related Works

The development of metallographic analysis is similar to that of general object detection. Nowadays, image processing methods of analyzing metallographic images are the best, because of their repeatability and consistency in digital image, and have been more accurate and faster than human judgement [6]. Petersen et al. [7] also explore a textural approach, viz. variance and range textural operators for ore type characterization and surface particle size estimation. Codaro et al. [8] used a method based on image processing that can be used for pits examination and classification in pitting corrosion evaluation. Lee et al. [9] used digital image analysis technique to automatically detect and segment the particle cracks in wrought aluminum alloys. Han et al. [10] proposed a deep learning method based on Inception-ResNet-v2 network for the industrial manufacturing defect defection field. AlexNet architecture was found to be a promising solution for the detection of specific geometric features in materials images [11]. Lin et al. [12] proposed the 3D convolutional neural network for segmenting casting defect regions.

The recent surge of interest in deep learning methods [13,14] is due to the fact that they have been shown to outperform previous state-of-the-art techniques in several tasks, as well as the abundance of complex data from different sources (e.g., visual, audio, medical, social, and sensor). Generally speaking, the field of object recognition has been dominated by CNN-based algorithms, which can be roughly divided into two-stage approaches and one-stage approaches. Based on the two-stage method, Girshick et al. [1] propose an object detector which applies a CNN to extract features for proposals generated by selective search [15]. The authors of the method call it regions with convolutional neural network (R-CNN). In the R-CNN family, model architectures are all region-based. Detection occurs in two stages: the first stage identifies a manageable number of bounding-box object region candidates. The second stage extracts CNN features from each region independently for classification. In 2015 [16], Girshick improved on R-CNN to build a faster object detector called Fast R-CNN, which is significantly faster in training and testing sessions over R-CNN. However, selective search is a time-consuming process that degrades network performance, so Shaoqing Ren et al. [17] propose Faster R-CNN, which eliminates selective search and lets the network learn the region proposals. The Mask R-CNN model introduced in 2017 [18], the most recent variation of the R-CNN family, models and supports both object detection and object segmentation. The R-CNN family may be among the most effective for object detection, achieving state-of-the-art results on computer vision benchmark datasets. Although accurate, the models can be slow when making predictions as compared to alternate models such as one-stage methods, which are designed for real-time prediction with less accuracy. In the following paper, we focus on model architectures which directly predict object bounding boxes for an image using a one-stage approach. The single shot multibox detector (SSD) [2] reaches new records in terms of performance and precision for object detection tasks. It is the first one-stage detector to achieve an accuracy reasonably close to two-stage detectors while still retaining the ability to work in real-time. The main advantage of this network is that it is fast, with reasonably good accuracy.

The YOLO (you only look once) model [3] is the first effort towards creating a fast real-time object detector. In 2017, the author of the previous paper [3] proposed YOLOv2 in an effort to further improve model performance in [19]. However, the YOLOv2 architecture still lacks important elements (residual blocks, shortcut connections, and upsampling) that are staples of most state-of-the art algorithms. YOLOv4 [5] incorporates all of these. It uses a custom deep architecture—darknet-53—with shortcut connections, upsampling, and concatenation, and it detects at three different scales. Although the YOLOv4 detector detects medium and large objects in images, it still fails to detect small objects in electron micrograph. Moreover, it is more accurate than YOLOv2, despite the larger time cost. Fortunately, the author released a lite version called Tiny YOLOv4, which uses a lighter model with fewer layers.

Though one-stage object detection algorithms such as SSD and YOLO detect objects in some detection areas quickly and accurately, these methods usually do not achieve good performance in optical micrographs. The main reason is that these methods perform poorly at detecting small objects in optical micrographs. After the celebrated victory of AlexNet [20] in the LSVRC2012 classification contest, the deep residual network (ResNet) [4] is currently the most outstanding work in the computer vision or deep learning community. Taking advantage of its powerful representational ability has boosted the performance of computer vision applications other than image classification, such as object detection and face recognition. The core idea of ResNet is that every additional layer should contain the identity function as one of its elements. Then, the depth level of ResNet is further expanded to 50 layers, 101 layers, and 152 layers. ResNet-X is a residual deep neural network with X layers; for example, ResNet-101 refers to a Resnet constructed using 101 layers. Residual connections enable the parameter gradients to propagate more easily from the output layer to earlier layers of the network, which makes it possible to train deeper networks. This increases network depth, which results in higher accuracies on more difficult tasks. He et al. [4] show that ResNet with 50/101/152 layers is more accurate than plain 18/34-layer networks. The strong performance of ResNet on image recognition and localization tasks show that depth is of central importance for many visual recognition tasks. However, in comparison to YOLOv4 [5], for object detection ResNet-152 yields similar performance, but shows poorer accuracy than YOLOv4 when finding small objects. For small object detection, YOLOv4 and SSD are better able to detect small objects when using a multi-scale model. Therefore, we propose a multi-scale Resnet (M-ResNet) detector especially designed for small object detection. Note that we employ dilated convolution at three different scales with different dilation rates to adapt the receptive fields for objects of different scales.

To enable robust defect detection, we need many pictures from the different metallic classes in order to train the computer to essentially convert pixel numbers to symbols. Nevertheless, to the best of our knowledge, there is no proper metallographic benchmark dataset for defect detection of metallography. The American Society for Metals (ASM) micrograph database [21] is a growing online collection of more than 4100 metallographic micrographs. However, the ASM micrographic database is not suitable for deep learning-based defect detection due to the micrographs of undamaged metallic structures shown in the dataset. Thus, we collect the metallographic database from the metallographic analysis laboratory of the Metal Industries Research and Development Centre (MIRDC). All of these metallographs are collected using the Zeiss Axiovert 200 Mat optical microscope, as shown in Figure 1.

## 3. Multi-Scale ResNet for Defect Detection

This section describes the proposed framework for metallographic analysis. We briefly review the powerful ResNet framework and present the proposed multi-scale ResNet detector. Finally, we discuss the retraining policy.

### 3.1. Deep Residual Network

ResNet [4], one of the most successful architectures in image classification, provides shortcut connections that allow a signal to bypass one layer and move to the next layer in the sequence. It allows for very deep networks of up to 152 layers by learning the residual representation functions instead of learning the signal representation directly. In actuality, with the shortcut connections, a linear shortcut is added to link the output of a residual block to its input thus enabling the flow of the gradient directly through these connections, which makes training a CNN much easier by reducing the vanishing gradient effect.

### 3.2. Architecture of Multi-Scale Resnets

In general, we define a small object as an object whose size is less than 1% of the total image area. In the ResNet architecture, the representation of fine details for small objects is totally lost in the coarse, semantic deepest layer. Therefore, in this work we attempt to detect objects of various sizes by making full use of residual blocks with rich fine details, increasing spatial resolution by dilated convolution and upsampling, and by merging using concatenation.

The proposed framework for metallographic analysis is shown in Table 1. Unlike the original ResNet-50, M-ResNet-50 predicts boxes at three different scales. That is, M-ResNet divides the input image using three different grid sizes to detect small, medium, and large objects, respectively. Take M-ResNet-50, for example: the M-ResNet-50 network uses ResNet-50, a custom variant of the ResNet architecture with a 50 layer network. For object detection, 50 more layers are stacked on top, yielding a 154-layer fully convolutional architecture as the basis for M-ResNet-50. In Table 1, it contains the feature extractor (Conv1–Conv5) and the detector. The feature extractor starts with a standard convolutional layer with 64 filters of size 3 × 3. To improve the accuracy of object detection and detect small objects, M-ResNet-50 is implemented with a feature pyramid network (FPN) [22]. FPN is a feature extractor designed around the pyramid concept with accuracy and speed in mind. It replaces the feature extraction of detectors such as Faster R-CNN and generates multiple feature map layers (multi-scale feature maps) with better quality information than the regular feature pyramid for object detection, especially for small objects.

In the following subsections, we discuss in detail the M-ResNet components: upsampling, concatenation, bottleneck residual blocks, feature pyramid networks, and dilated convolution.

#### 3.2.1. Upsampling and Concatenation

Detection at different layers helps to facilitate the detection of small objects, which is a known weakness of ResNet. In the M-ResNet architecture, upsampled layers concatenate with previous layers, helping to preserve the fine-grained features which help in detecting objects of various sizes. Table 1 shows the concatenation of the feature maps of the upper layers with the lower layer features through upsampling layers. In addition, low-layer features that possess limited representation power are enriched by the concatenation of higher-layer features, resulting in good representation power for small object detection without incurring significant additional computational costs.

#### 3.2.2. Bottleneck Residual Blocks

As shown in Figure 2a,b, a bottleneck residual network is similar to the original residual network. In the ResNet-50 architecture [4], a bottleneck architecture is used to reduce computational time. The bottleneck residual networks are composed of a sequence of three convolutional layers with filters of size 1 × 1, 3 × 3, and 1 × 1, respectively. This amounts to using a 1 × 1 convolution to reduce the channels of the input before performing the expensive 3 × 3 convolution, and then using another 1 × 1 convolution to project it back into the original shape. When the input and output feature maps are of the same dimensions, the shortcut connections can be used directly. In the M-ResNet architecture, the main features of metallographic images are extracted by the dilated residual network (described in Section 3.2.4).

#### 3.2.3. Feature Pyramid Network

The feature pyramid network, proposed by T.-Y. Lin et al. [22], is used in work on object detection or image segmentation. In contrast to other detection methods, FPN detects multiscale objects by upsampling the feature maps of the CNN feature extractor, resulting in considerable performance gains in multi-scale object detection, as shown in Figure 3. Because higher-resolution feature maps are used for smaller-scale object detection, many FPN variants enrich those features. This feature pyramid has high resolution to capture fine structures, sufficiently rich semantics to accurately predict class labels and capture multi-scale information to predict the original images at all levels, and is built quickly from a single input image scale. Therefore, to achieve accurate predictions, FPN is used for this work.

#### 3.2.4. Dilated Convolution

To increase feature resolution, which is necessary to generate high quality results, recent top methods [23,24] rely heavily on the use of dilated convolution [25]. Dilated convolution decreases computational costs by adding dilation rate to the conv2D kernel. The dilation rate is the spacing between each pixel in the convolutional filter. A 3 × 3 kernel with a dilation rate of 2 has the same field of view as a 5 × 5 kernel. Increasing the field of view has the added advantage of increasing the receptive field, and thus helps the filter to capture more contextual information. For large objects, M-ResNet uses a larger dilation rate to increase the receptive field. In contrast, we use a smaller dilation rate for small objects. Moreover, by detecting details at higher resolutions, we detect finer details in the images. As seen in Figure 4, dilated convolution enlarges the receptive field of a convolutional kernel without increasing the number of parameters. In FPN feature extraction, we adopt a dilated convolution with rates 1, 2, and 3 to extract features from the original images. By comparing with the normal convolutional kernel, the dilated convolutional kernel gleans more information from the extended receptive field.

### 3.3. Retraining Strategy

Modern object detection models have millions of parameters. Training them using the M-ResNet architecture requires many labeled training data and much computing power. Transfer learning is a technique that shortcuts much of this by taking a piece of a model that has already been trained on a related task and reusing it in a new model. In this paper, we propose a retraining strategy that promotes high performance in object detection. In general, there are never enough training images for object detection models, due to a lack of object samples or because of the cumbersome image labeling process. Therefore, in each testing phase, we use the results of testing to enhance the robustness of the proposed model, as shown in Figure 5. For poor results, we compare with previously labeled images and correct them to implement retraining. Automated pre-labeling employs the last models to label the test images automatically, saving much manual labeling time. The M-ResNet models achieve outstanding performance for object detection through the use of retraining strategies that cater to the final application.

## 4. Experimental Results and Analysis

In this section, we evaluate M-ResNet on MIRDC metallographic dataset. All the experiments are implemented in CUDA C++ API on a machine with NVIDIA 1080Ti GPUs. We use the well-trained ResNet-50/101/152 models as the pre-trained model for M-ResNet training, and then fine-tuned the model on the MIRDC metallographic dataset. The performance is measured by average precision (AP) [26]. AP is a measure that combines recall and precision for ranked retrieval results. We compared the results with the state-of-the-art CNN in terms of AP and inference speed. The input metallographs were resized to 736×416 pixels. For all experiments, the weight decay was set to 0.0005, the momentum to 0.9, and the learning rate to 0.001. The overlap threshold for a region of interest (ROI) to be considered as foreground is 0.5, and the number of ROIs per batch is 32.

### 4.1. Metallographic Dataset

There are currently no appropriate metallographic datasets for deep learning. Thus, we collected the metallographic images from metallographic analysis laboratory of MIRDC. The metallographic datasets contain five different classes of metallographs under the optical microscopy 100× magnification, such as carbon steel, medium carbon steel, medium carbon Cr-Mo alloy steel, chrome vanadium steel and chromium-molybdenum steel, as shown in Figure 6. Each class varies from 100 to 200 metallographs and contains a different number of normal and defective images. Therefore, the metallographic dataset is composed of 818 images in the 24-bit JPEG format. In the metallographic dataset, we note that there are far fewer positives (defective images) than negatives (normal images).

It is a generally accepted notion that bigger datasets result in better deep learning models [27]. Therefore, we can improve the performance of the model by augmenting the data. Image data augmentation is a technique that can be used to artificially expand the size of a training dataset by creating modified versions of images in the dataset. In addition, this data augmentation helped reduce overfitting when training a deep neural network. In Figure 7, we used the most common three main policies of data augmentation techniques for image data, such as flipping, rotation, and zooming. Each policy consists of three associated parameters. Thus, in this study, the data augmentation with 9 operations is used to train the models for each image. In total, we used 8180 metallographs to evaluate our proposed method, including 6261 normal images and 1919 defective images. We divided the metallographic datasets into training and testing sets by randomly splitting the dataset. In our experiments, we used 80% of the dataset for training, and the remaining 20% for testing.

### 4.2. Experimental Results and Analysis

During testing, predicted boxes with confidence scores greater than 0.5 were regarded as defective areas of metallographic images. The proposed M-ResNet framework has been implemented by a window-based program shown in Figure 8. Based on the concept of “What You See Is What You Get” (WYSIWYG), the window consists of two parts, which are called the parameter setting and the main image windows, respectively. The user can explore the detection result of the entire image in the main image window. There are five different classes in metallographic datasets, so the program includes five different models for metallographic analysis. Therefore, the user can easily change Cls (name of class), Cfg (configuration file), and Wei (model weights) to find the most suitable model. By clicking the run button, the program will execute batch processing of images after the parameters are determined. In Figure 9, the window shows several selected metallographs outputted from the program. In recent years, some research papers [28,29,30] on object detection based on deep learning methods also use the sized category of AP_XS_, AP_S_, AP_M_, AP_L_, for the evaluation of the effectiveness of the proposed methods. The size of the defective area is defined by the number of pixels in the defective area of the metallograph. We assigned each defective object to a size category depending on the percent of the defective area in the metallographic image. In this work, the sizes of defective areas were divided into extra small (XS: under 0.3%), small (S: 0.3–1%), medium (M: 1–5%), and large (L: above 5%).

Table 2 shows our detection results on the MIRDC dataset for various ResNet architectures, including AP_XS_, AP_S_, AP_M_, AP_L_ (AP at different scales: extra-small, small, medium, and large), mean average precision (mAP), and computational costs. The results show that the proposed M-ResNet architectures with the MIRDC dataset achieve better mAPs. For instance, M-ResNet-50 achieves better performance than ResNet-50, improving the mAP from 65.1% to 78.5%. In addition, we also show the results for the different categories to demonstrate the strong performance for small-object detection. For single-object detection with extra-small objects, the proposed method (M-ResNet-50/101/152) yields APs of 78.5%, 81.4%, and 85.7%, respectively. Among the ResNet architectures, ResNet-152/101/50 models are faster than M-ResNet-152/101/50 models in computational costs. M-ResNet-152 is the slowest in this regard. Unfortunately, the M-ResNet-152/101/50 models contain more layers than the original ResNet-152/101/50, leading to greater computational costs. This is because multi-scale architectures that make use of upsampling, concatenation, and FPN as in M-ResNet increase computational costs.

### 4.3. Comparison with Other Object Detectors

As YOLOv4 is a state-of-the-art object detection algorithm, we trained YOLOv4 on MIRDC dataset and compared the results with that of the proposed method. YOLOv4 yielded a mAP of 78.1%. In addition, we also compared our method with other well-known algorithms, as shown in Table 3. M-ResNet-152 clearly outperforms previous deep learning algorithms such as SSD300, Faster R-CNN, and YOLOv4. It also significantly improves the accuracy of one-stage object detection, e.g., the accuracy of M-ResNet-152 increases by 11% and 7.6% compared to the state-of-the-art one-stage detectors SSD300 and YOLOv4, respectively. The main reason is that these two detectors are less sensitive to small objects than M-ResNet-152. However, the M-ResNet-152/101/50 mAP results in Table 3 suggest that these performance gains come at added computational expense: we note the large speed differences between the various methods. In Table 3, ResNet-50 is among the fastest models in our experiments. Faster R-CNN is slowest in this regard. Unfortunately, the M-ResNet-152/101/50 models contain more layers than SSD300, YOLOv4, and the original ResNet-152/101/50, leading to greater computational costs. Nevertheless, the M-ResNet-50 model is still useful for real-time object detection.

In these experiments, some defective objects are still not accurately detected by the proposed algorithm. Analysis of the testing results reveals that some objects are hidden, or are too tiny, and that some kind of defective object shapes are not part of training. It is quite a challenging job to solve these problems for future research. Automatically recognizing defective objects is essential for handling metallographs. We hope to further improve the classifier using more advanced model ablation and auxiliary methods, to facilitate the accurate recognition of metallographs when scientists use these methods to analyze metallography.

## 5. Conclusions

In this work, we propose a new deep learning method and apply it to metallographic analysis. In addition to normal bottleneck ResNet networks, we also employ the state-of-the-art FPN and dilated ResNet networks in the M-ResNet architecture. This improvement enlarges the receptive field in multi-scale feature maps and extracts more object information. In addition, we present a new dataset to evaluate the effectiveness of M-ResNet for metallographic analysis. Most previous methods are ill-suited to metallographic applications, due to low mAPs and slow computing speed. Fortunately, for the defect detection of metallographic analysis, deep learning methods greatly improve detection accuracy. As illustrated in the Table 3, for the MIRDC dataset, the APs of Faster R-CNN, SSD300, YOLOv4 and the proposed M-ResNet-152 are 77.8%, 74.1%, 78.1%, and 85.7%, respectively, which indicates that deep learning is to play an increasingly important role in defect detection in metallographic analysis. On the whole, the results of this research are not outstanding, but inspire a novel way of thinking with metallographic analysis. In future work, we will compile a larger dataset based on generative adversarial networks (GANs) [31,32] and further improve the algorithm to boost detection accuracy. We believe that M-ResNet will find use in many successful metallographic applications.

## Figures and Tables

**Figure 1 sensors-20-05593-f001:**
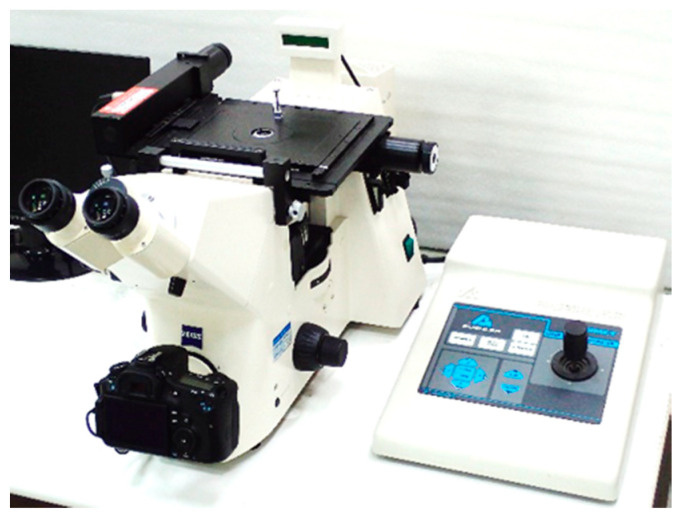
Zeiss Axiovert 200 Mat optical microscope.

**Figure 2 sensors-20-05593-f002:**
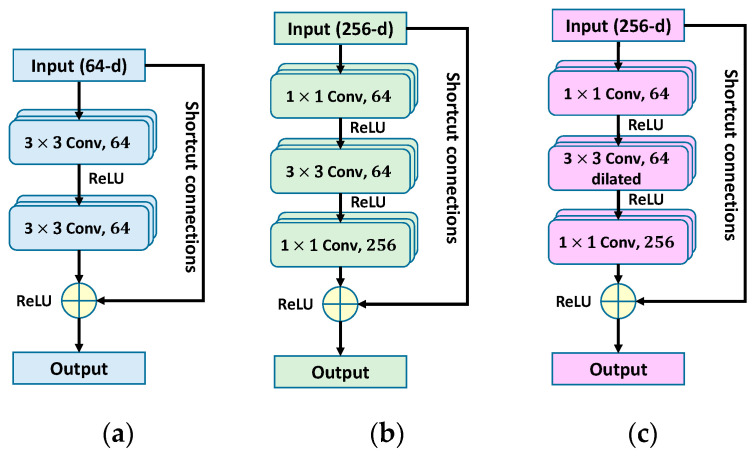
(**a**) Basic block of residual network; (**b**) bottleneck residual network; (**c**) dilated residual network.

**Figure 3 sensors-20-05593-f003:**
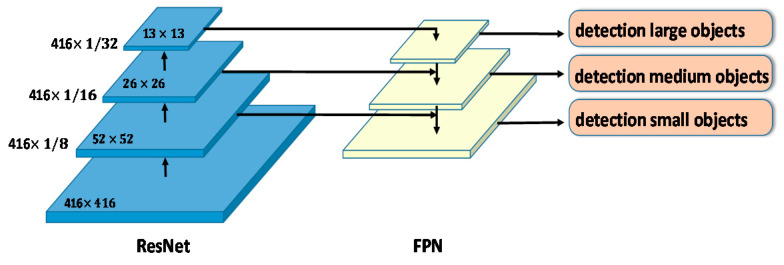
Framework of ResNet and feature pyramid network (FPN).

**Figure 4 sensors-20-05593-f004:**
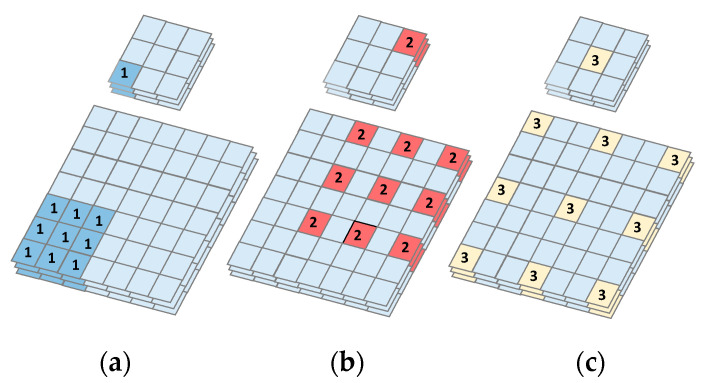
A 3D example of dilated convolution. Convolution layer with kernel size 3 × 3, (**a**) normal dilated convolution with rate = 1; (**b**) dilated convolution with rate = 2; (**c**) dilated convolution with rate = 3.

**Figure 5 sensors-20-05593-f005:**
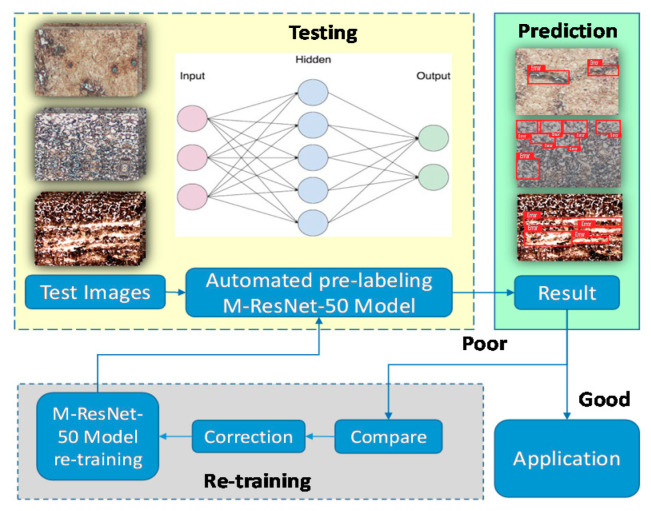
M-ResNet retraining strategy.

**Figure 6 sensors-20-05593-f006:**
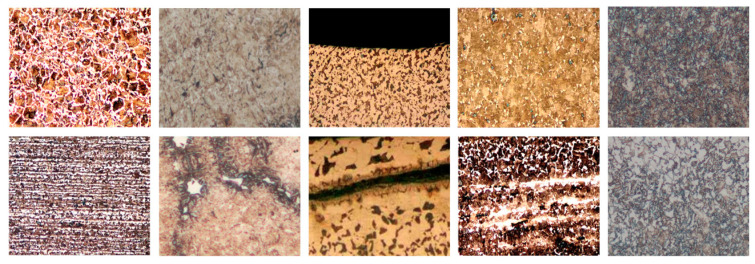
Five different classes of metallographic images differentiated form left to right. The top rows are normal images, and bottom rows are defective images.

**Figure 7 sensors-20-05593-f007:**
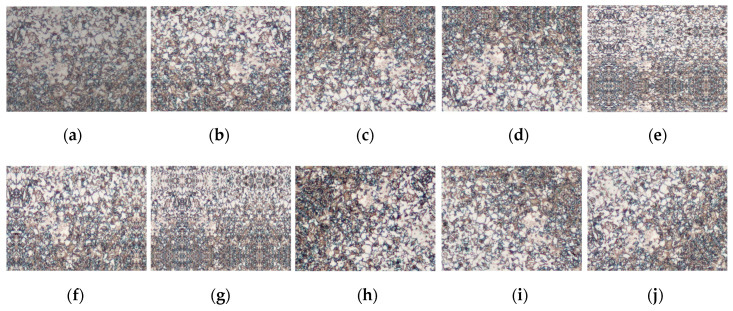
Examples of data augmentations with 9 operations that can be used to train the models for each image. (**a**) original, (**b**–**d**) flipping, (**e**–**g**) zooming, (**h**–**j**) rotation.

**Figure 8 sensors-20-05593-f008:**
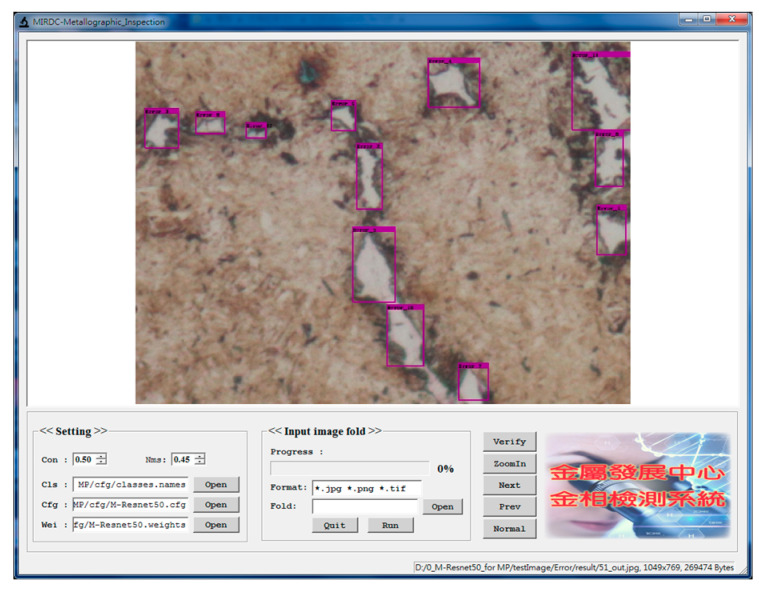
The window-based program used to implement the proposed M-ResNet for metallographic analysis.

**Figure 9 sensors-20-05593-f009:**
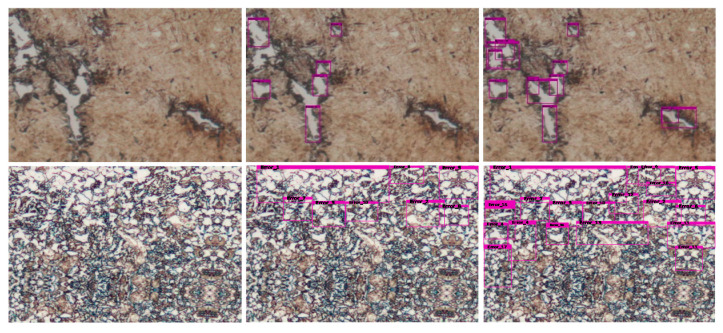
Selected examples of defect detection results on the Metal Industries Research and Development Centre (MIRDC) dataset for various methods. Left column is original images, and middle and right columns are results of ResNet-50 and M-ResNet-50 models.

**Table 1 sensors-20-05593-t001:** M-Resnet-50 architecture.

Layer	Operation Type	Input (Pixel)	Filter	Stride	Dilation	Output (Pixel)	
0	Convolution	416×416×3	64	2	−	208×208×64	Conv1
1	Max pooling	208×208×64	−	2	−	104×104×64	
2–5	Bottleneck residual network Network	104×104×64	256	1	−	104×104×256	Conv2
6–9	Bottleneck residual network	104×104×256	−	1	−	104×104×256	
10–13	Bottleneck residual network	104×104×256	−	1	−	104×104×256	
14–17	Bottleneck residual network	104×104×256	512	2	−	52×52×512	Conv3
18–21	Bottleneck residual network	52×52×512	−	1	−	52×52×512	
22–25	Bottleneck residual network	52×52×512	−	1	−	52×52×512	
26–29	Bottleneck residual network	52×52×512	−	1	−	52×52×512	
30–33	Dilated residual network	52×52×512	1024	2	3	26×26×1024	Conv4
34–37	Dilated residual network	26×26×1024	−	1	3	26×26×1024	
38–41	Dilated residual network	26×26×1024	−	1	3	26×26×1024	
42–45	Dilated residual network	26×26×1024	−	1	3	26×26×1024	
46–49	Dilated residual network	26×26×1024	−	1	3	26×26×1024	
50–53	Dilated residual network	26×26×1024	−	1	3	26×26×1024	
54–57	Dilated residual network	26×26×1024	2048	2	3	13×13×2048	Conv5
58–61	Dilated residual network	13×13×2048	−	1	3	13×13×2048	
62–65	Dilated residual network	13×13×2048	−	1	3	13×13×2048	
66	Convolution	13×13×2048	2048	1	−	13×13×2048	
67	Convolution	13×13×2048	18	1	−	13×13×18	
68	Large-object detection						
69	Route	29					
70–73	Dilated residual network	52×52×512	1024	2	2	26×26×1024	Conv4
74–77	Dilated residual network	26×26×1024	−	1	2	26×26×1024	
78–81	Dilated residual network	26×26×1024	−	1	2	26×26×1024	
82–85	Dilated residual network	26×26×1024	−	1	2	26×26×1024	
86–89	Dilated residual network	26×26×1024	−	1	2	26×26×1024	
90–93	Dilated residual network	26×26×1024	−	1	2	26×26×1024	
94–97	Dilated residual network	26×26×1024	2048	2	2	13×13×2048	Conv5
98–101	Dilated residual network	13×13×2048	−	1	2	13×13×2048	
102–105	Dilated residual network	13×13×2048	−	1	2	13×13×2048	
106	Convolution	13×13×2048	1024	1	−	13×13×1024	
107	2 × Upsampling	13×13×1024	1024			26×26×1024	
108	Concatenation	107, 33					
109	Convolution	26×26×2048	1024	1	−	26×26×1024	
110	Convolution	26×26×1024	18	1	−	26×26×18	
111	Medium-object detection						
112	Route	29					
113–116	Dilated residual network	52×52×512	1024	2	1	26×26×1024	Conv4
117–120	Dilated residual network	26×26×1024	−	1	1	26×26×1024	
121–124	Dilated residual network	26×26×1024	−	1	1	26×26×1024	
125–128	Dilated residual network	26×26×1024	−	1	1	26×26×1024	
129–132	Dilated residual network	26×26×1024	−	1	1	26×26×1024	
133–136	Dilated residual network	26×26×1024	−	1	1	26×26×1024	
137–140	Dilated residual network	26×26×1024	2048	2	1	13×13×2048	Conv5
141–144	Dilated residual network	13×13×2048	−	1	1	13×13×2048	
145–148	Dilated residual network	13×13×2048	−	1	1	13×13×2048	
149	Convolution	13×13×2048	512	1	−	13×13×512	
150	4 × Upsampling	13×13×512	512			52×52×512	
151	Concatenation	150, 17					
152	Convolution	52×52×1024	512	1	−	52×52×512	
153	Convolution	52×52×512	18	1	−	52×52×18	
154	Small-object detection						

**Table 2 sensors-20-05593-t002:** Detection results under various ResNet architectures.

Dataset	Method	AP_XS_ (%)	AP_S_ (%)	AP_M_ (%)	AP_L_ (%)	mAP (%)	Speed (FPS)
MIRDC dataset	M-ResNet-50	67.5	78.7	83.1	84.7	78.5	45
	M-ResNet-101	73.4	80.1	85.4	86.7	81.4	32
	M-ResNet-152	78.9	85.4	88.3	90.2	85.7	27
	ResNet-50	55.8	60.7	68.1	75.7	65.1	87
	ResNet-101	62.5	67.1	72.4	78.4	70.1	58
	ResNet-152	69.7	73.4	76.3	82.2	75.4	44

**Table 3 sensors-20-05593-t003:** Comparison between object detectors.

Method	Backbone Network	mAP (%)	Speed (FPS)
Faster R-CNN	ResNet-101	77.8	-
SSD300	VGG-16	74.7	48
YOLOv4	darknet53	78.1	47
ResNet-50	ResNet-50	65.1	87
M-ResNet-50	ResNet-50	78.5	45
ResNet-101	ResNet-101	70.1	58
M-ResNet-101	ResNet-101	81.4	32
ResNet-152	ResNet-152	75.4	44
M-ResNet-152	ResNet-152	85.7	27

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
