# Peer review of "A Study of Defect Detection Techniques for Metallographic Images"

_sensors, 2020, doi:10.3390/s20195593_

Round 1
Reviewer 1 Report
This article present Defect Detection Techniques for Metallographic Images by automatic detection. Deep learning has been applied to different problems in computer vision in this study, authors propose a novel convolutional neural network architecture for metallographic analysis based on a modified residual neural network (ResNet). Multi-scale ResNet (M-ResNet.
The article is interesting, but what happens when real word metallographic experimental images was taking into account, not only data base images. Do the authors think that is an important issue, why or why not should be included in abstract, discussion and conclusions.
Author Response
Response to the suggestions of Reviewer 1:
This article present Defect Detection Techniques for Metallographic Images by automatic detection. Deep learning has been applied to different problems in computer vision in this study, authors propose a novel convolutional neural network architecture for metallographic analysis based on a modified residual neural network (ResNet). Multi-scale ResNet (M-ResNet).
The article is interesting, but what happens when real word metallographic experimental images was taking into account, not only data base images. Do the authors think that is an important issue, why or why not should be included in abstract, discussion and conclusions.
Response : We thank the reviewer for the constructive suggestions. Actually, each metallographic image in our database is the real-world image. We collected the metallographic database from metallographic analysis laboratory of Metal Industries Research and Development Centre (MIRDC). All of these metallographes are collected using the Zeiss Axiovert 200 Mat optical microscopy, as shown in Figure 1. The discussion of the issue is described to the line 127-130 of the manuscript.

Reviewer 2 Report
This research work focus in detecting small defects in metallographic images.
The related work have neither adequate nor recent references. I suggest the authors to add more recent references.
The authors explained the used dataset in section 4.1 that includes five classes of 818 metallographic images. They used the common data augmentation, I suggest the authors to use GAN-based data augmentation to improve and balance their dataset.
In experimental section, the authors assigned defects based on the size of defective area to a size-category. However, their is no any discussion related with the size-category. I suggest the authors to explain who the proposed algorithm succeed/failed to detect any of the size-category defects. I mean the mAP to detect each size-category.
Author Response
This research work focus in detecting small defects in metallographic images.
Point 1: The related work has neither adequate nor recent references. I suggest the authors to add more recent references.
Response 1: Thank you for the constructive suggestions. Yes, I completely agree with the reviewer’s opinions. The discussion with other competing methods can promote our work. Therefore, several recent works about the metallographic images defect detection methods are provided and described in the line 62-66 of the manuscript.
Point 2: The authors explained the used dataset in section 4.1 that includes five classes of 818 metallographic images. They used the common data augmentation, I suggest the authors to use GAN-based data augmentation to improve and balance their dataset.
Response 2: We would like to express our appreciation for the effort Reviewer has made on this paper to help us improve the paper’s quality. By using the Generative Adversarial Networks (GAN) to generate artificial training data for machine learning tasks, the generation of artificial training data can be extremely useful in situations such as imbalanced data sets. Your valuable comments definitely can promote the performance of our system. However, due to the inevitable deadline of the revised paper which should be completely upload within 10 days, we do not have enough time to make this experiment. We sincerely appreciate all your constructive suggestions. We are looking to finish this experiment in the future.
Point 3: In experimental section, the authors assigned defects based on the size of defective area to a size-category. However, there is no any discussion related with the size-category. I suggest the authors to explain who the proposed algorithm succeed/failed to detect any of the size-category defects. I mean the mAP to detect each size-category.
Response 3: Thank you for the valuable suggestions. Because of the paper focuses on Multi-scale ResNet (M-ResNet) methods for defect detection, we verify the performance of M-ResNet for various sizes. In recent years, some research papers [26-28] of object detection based on deep learning method are also used by the size category of APXS, APS, APM, APL for evaluation of the effectiveness of proposed methods (The sentences are added to the line 281-283 of the manuscript), which can be seen in the following papers.
- X Long, K Deng, G Wang, Y Zhang, Q Dang, Y Gao, H Shen, J Ren, “PP-YOLO: An Effective and Efficient Implementation of Object Detector” arXiv preprint arXiv:2007.12099, 2020. (Page 6, Table 1)
- Licheng Jiao, Fan Zhang, Fang Liu, Shuyuan Yang, Lingling Li, Zhixi Feng, Rong Qu, “A Survey of Deep Learning-Based Object Detection”, IEEE Access, Vol. 7, pp. 128837 – 128868, 2019. (Table 1)
- Shunjun Wei, Hao Su, Jing Ming, Chen Wang, Min Yan, Durga Kumar, Jun Shi and Xiaoling Zhang, “Precise and Robust Ship Detection for High-Resolution SAR Imagery Based on HR-SDNet”, Remote Sensing, Vol. 12, issue 1, p. 167, 2020. (Table 3)

Reviewer 3 Report
In this paper, a method based on image analysis technique for the automatic identification of defects on metallographic analysis is introduced. The method yields on deep learning with the innovation of introducing a multi-scale residual neuronal network in the method.
The paper is well written (few specific comments are report below), methods, analyses and results are generally well presented, and the findings are interesting and, even if not outstanding and not completely solving the issue of expert analysis, potentially relevant. Nevertheless, in few points, the quality of the discussion could be improved and the impact of the paper and its readability for the audience of Sensors would be consequently higher. The reviewer thinks this paper will be of sure interest for that audience, even if some revisions are needed to improve its impact. General comments and specific comments are given below.
GENERAL COMMENTS
- From page 4 onward, also on table 1, often the unit of measure of digital image (or of a portion of it) are missing; please add px where appropriate.
- On figure 5, the cycle generating when the result is poor is potentially infinite: can THE Authors explain what changes at every cycle and how do they solve this? Is there a maximum number of cycles before to exit? What happened when this was tested on the real images? Please explain deeper.
- How do the Authors define and compute precision (page 8)?
- To the reviewer is not clear what is shown on figure 7: is always the same image with different data augmentation technique applied? If so, can this be explained in the text and (briefly) in the caption?
- On table 2, similarly to table 3, a column with the computational time (or speed in FPS) should be added, in order to understand if there is a difference among the various scales. If so, this could be discussed in the main text.
SPECIFIC COMMENTS
- Page 1 line 37: the comma after “cases” should be deleted
- Page 2 line 678: the t in “…stages: The first stage…” should be lower case
- Page 2 line 72: “However, selective search is a time-consuming process that degrades network performance,” SO “Shaoqing Ren et al. [12] propose Faster R-CNN,…”
- Page 3 line 108: “He et al. show” not shows
- Page 3 line 120: explain the acronym ASM and add a link to the dataset
- Page 4 line 168: the font of “Bott” is different from the rest of the paper
- Page 8 line 231: MIRDC should be explained here and not later on line 242-243
- Page 8 line 256: common instead of commonly
- The format used in the References is not uniform
Author Response
In this paper, a method based on image analysis technique for the automatic identification of defects on metallographic analysis is introduced. The method yields on deep learning with the innovation of introducing a multi-scale residual neuronal network in the method.
The paper is well written (few specific comments are report below), methods, analyses and results are generally well presented, and the findings are interesting and, even if not outstanding and not completely solving the issue of expert analysis, potentially relevant. Nevertheless, in few points, the quality of the discussion could be improved and the impact of the paper and its readability for the audience of Sensors would be consequently higher. The reviewer thinks this paper will be of sure interest for that audience, even if some revisions are needed to improve its impact. General comments and specific comments are given below.
GENERAL COMMENTS
Point 1: From page 4 onward, also on table 1, often the unit of measure of digital image (or of a portion of it) are missing; please add px where appropriate.
Response 1: We would like to express our appreciation for the effort reviewer has made on this paper to help us improve the paper’s quality. We added the “pixel” in Table 1.
Point 2: On figure 5, the cycle generating when the result is poor is potentially infinite: can THE authors explain what changes at every cycle and how do they solve this? Is there a maximum number of cycles before to exit? What happened when this was tested on the real images? Please explain deeper.
Response 2: Thank you for the constructive suggestions. Deep learning and other modern nonlinear machine learning techniques get better with more data. Moreover, in deep learning, the quality of models is generally constrained by the quality of training data. Therefore, in each testing phase, we use the results of testing to enhance the robustness of the proposed model. For poor results, we compare with previously labeled images and correct them to implement retraining. Constantly correcting labeled images errors by manual can increase the quality of training data. Therefore, it is not going to happen that the cycle generating when the result is poor is potentially infinite. Actually, each metallographic image in our database is the real image. We collect the metallographic database from metallographic analysis laboratory of Metal Industries Research and Development Centre (MIRDC). All of these metallographes are collected using the Zeiss Axiovert 200 Mat optical microscopy, as shown in Figure 1. In our retraining strategy, we also can promote the robust of our model by the real images.
Point 3: How do the Authors define and compute precision (page 8)?
Response 3: Thank you for the constructive suggestions. Yes, I completely agree with the reviewer’s opinions. In our manuscript, we used the metric AP (average precision) and mAP (mean average precision) to compare the performance of our systems. Average precision is a measure that combines recall and precision for ranked retrieval results. They are from the Everingham et. al. [24] - 4.2 Evaluation of Results (Page 11). AP is a popular metric in measuring the accuracy of object detectors like Faster R-CNN, SSD, etc. AP computes the average precision value for recall value over 0 to 1. In addition, mAP (mean average precision) is the average of AP. The define of AP and mAP are provided and described in the line 239-240.
Point 4: To the reviewer is not clear what is shown on figure 7: is always the same image with different data augmentation technique applied? If so, can this be explained in the text and (briefly) in the caption?
Response 4: I'm sorry to cause you misunderstanding. The explanations of figure 7 are described in the line 268-269.
Point 5: On table 2, similarly to table 3, a column with the computational time (or speed in FPS) should be added, in order to understand if there is a difference among the various scales. If so, this could be discussed in the main text.
Response 5: Yes, I completely agree with the reviewer’s opinions. We added this column with Speed (FPS) in Table 2. Among the ResNet architectures, ResNet-152/101/50 models are faster than M-ResNet-152/101/50 models in computational costs. M-ResNet-152 is the slowest in this regard. Unfortunately, the M-ResNet-152/101/50 models contain more layers than the original ResNet-152/101/50, leading to greater computational costs. This is because multi-scale architectures that make use of upsampling, concatenation, and FPN as in M-ResNet increase computational costs. Some discussions about computational costs are described in line 297-302.
Point 6: SPECIFIC COMMENTS
Page 1 line 37: the comma after “cases” should be deleted
Page 2 line 67: the t in “…stages: The first stage…” should be lower case
Page 2 line 72: “However, selective search is a time-consuming process that degrades network performance,” SO “Shaoqing Ren et al. [12] propose Faster R-CNN,…”
Page 3 line 108: “He et al. show” not shows
Page 3 line 120: explain the acronym ASM and add a link to the dataset
Page 4 line 168: the font of “Bott” is different from the rest of the paper
Page 8 line 231: MIRDC should be explained here and not later on line 242-243
Page 8 line 256: common instead of commonly
The format used in the References is not uniform
Response 6: We would like to express our appreciation for the effort Reviewer has made on this paper to help us improve the paper’s quality. The English in our revision has been checked by two professional editors, both native speakers of English. All typos and grammatical errors have been corrected.

Round 2
Reviewer 3 Report
The reviewer greatly appreciate the improvements made by the Authors. All my points have been properly faced. In my opinion, the paper can be published in the present form.
Author Response
The reviewer greatly appreciate the improvements made by the Authors. All my points have been properly faced. In my opinion, the paper can be published in the present form.
Response: Thank you for your affirmation and encouragement.
